# Modeling and Optimization of the Culture Medium for Efficient 4′-N-Demethyl-Vicenistatin Production by *Streptomyces parvus* Using Response Surface Methodology and Artificial-Neural-Network-Genetic-Algorithm

**Zhixin Yu** [1]**, Hongxin Fu** [1,2,3,]*** and Jufang Wang** [1,2,3,]*** 

[1] School of Biology and Biological Engineering, South China University of Technology, Guangzhou 510006, China; nikuspud_minoyyy@foxmail.com

[2] Guangdong Provincial Key Laboratory of Fermentation and Enzyme Engineering, South China University of Technology, Guangzhou 510006, China

[3] State Key Laboratory of Pulp and Paper Engineering, South China University of Technology, Guangzhou 510006, China

* Correspondence: hongxinfu@scut.edu.cn (H.F.); jufwang@scut.edu.cn (J.W.)

**Abstract:** 4′-N-demethyl-vicenistatin is a vicenistatin analogue that has better antitumor activity with promising applications in the pharmaceuticals industry. The harnessing of the complete potential of this compound necessitates a systematic optimization of the culture medium to enable the cost-effective production of 4′-N-demethyl-vicenistatin by *Streptomyces parvus* SCSIO Mla-L010/Δ*vicG*. Therefore, in this study, a sequential approach was employed to screen the significant medium compositions, as follows: one-factor-at-a-time (OFAT) and Plackett–Burman designs (PBD) were initially utilized. Cassava starch, glycerol, and seawater salt were identified as the pivotal components influencing 4′-N-demethyl-vicenistatin production. To further investigate the direct and interactive effects of these key components, a three-factor, five-level central composite design (CCD) was implemented. Finally, response surface methodology (RSM) and an artificial-neural-network-genetic-algorithm (ANN-GA) were employed for the modeling and optimization of the medium components to enhance efficient 4′-N-demethyl-vicenistatin production. The ANN-GA model showed superior reliability, achieving the most 4′-N-demethyl-vicenistatin, at 0.1921 g/L, which was 17% and 283% higher than the RSM-optimized and initial medium approaches, respectively. This study represents pioneering work on statistically guided optimization strategies for enhancing 4′-N-demethyl-vicenistatin production through medium optimization.

**Keywords:** culture medium optimization; response surface methodology; 4′-N-demethyl-vicenistatin; *Streptomyces parvus*; artificial-neural-network-genetic-algorithm



## 1. Introduction

Antibiotics are a significant class of metabolites produced by microorganisms, animals, or plants. The biological activities of antibiotics, such as cytotoxicity, bacteriostatic, antimalarial, and anti-parasitic properties, make them widely used in medicine, agriculture, animal husbandry, the food industry, and other fields [1]. In recent years, the exploitation of antibiotic resources has gradually shifted from soil to the ocean [2]. Typically, *Streptomyces parvus* is considered as a potential source of biologically active compounds that has been explored widely for drug development [3]. For example, ZM-1 (identified as holomycin) was isolated from *S. parvus* 33 and found to have strong antibacterial activity against plant pathogenic fungi [4]. *S. parvus* NEAE-95 produced an anti-neoplastic agent, L-asparaginase, which was used in acute lymphoblastic leukemia treatment [5]. LYRM03, isolated from *S. parvus* HCCB10043, showed higher potent inhibitory activity against aminopeptidase N for cancer therapy than bestatin [6]. Eumelanin pigment, purified from *S. parvus* BSB49, could be utilized for pharmaceutic and

cosmetic product development [7]. Silver nitrate nanomaterials from *S. parvus* Al-Dhabi-91 were a suitable active substance for treating infectious disease [8].

Vicenistatin, a 20-membered macrolactam core with an amino-sugar vicenisamine, was first isolated from marine microorganisms, *S. halstedii* HC34, and showed great potential to be used as an antitumor drug. It not only exhibits cytotoxicity against COLO205 and HL-60 cells, but also has antitumor activity toward Co-3 cells [9]. Recently, a vicenistatin analogue (4′-N-demethyl-vicenistatin), which showed better antitumor activity and reduced cytotoxicity than vicenistatin, was isolated from *S. parvus* SCSIO Mla-L010/Δ*vicG* (a disruptant of the N-methyltransferase gene). It was characterized as an macrolactam antibiotic and impressive antitumor drug. In comparison to vicenistatin, 4′-N-demethyl-vicenistatin exhibited good antimicrobial activities, including methicillin-resistant *Staphylococcus aureus*, methicillin-resistant *Staphylococcus epidermidis*, *Micrococcus luteus*, and *Bacillus subtilis*, with low cytotoxicity [10]. However, the low product concentration of 4′-N-demethyl-vicenistatin (4 mg/L) was the main limiting factor for subsequent medicinal property evaluation and applications [11].

The compositions and concentration of a fermentation medium play a crucial role in the growth of microorganisms and the formation of secondary metabolites. Thus, it is essential to identify and optimize the important components (such as carbon sources, nitrogen sources, and inorganic salts) in the medium for efficient 4′-N-demethyl-vicenistatin production. For fermentation medium optimization, the one-factor-at-a-time (OFAT) method is a basic strategy and is frequently used, but it does not consider the interactions among various factors [12]. In contrast, response surface methodology (RSM) can elucidate the interactions among individual factors. Prior to employing RSM, the Plackett–Burman design (PBD) was often conducted to identify the factors that exert a significant influence on the outcomes [13,14]. Apart from the above classical methods, machine learning tools such as the artificial-neural-network-genetic-algorithm (ANN-GA) have been confirmed to possess a better predictive capability, especially for complex and nonlinear processes like biological fermentation [15,16]. However, there is no related report concerning the systematic study of the effects of medium compositions, nor the model for optimizing the fermentation medium to enhance 4′-N-demethyl-vicenistatin production.

The aim of this study is to optimize the fermentation medium in order to achieve efficient 4′-N-demethyl-vicenistatin production using *S. parvus* SCSIO Mla-L010/Δ*vicG* with the statistical design of experiments. First, the medium compositions were screened by OFAT and PBD. Then, the comparative performance of RSM and ANN-GA for modeling and optimizing the medium compositions was conducted. Finally, the predictive ability of RSM and ANN-GA was experimentally confirmed and achieved a 226% and 283% improvement in 4′-N-demethyl-vicenistatin production, respectively.

## 2. Results

### 2.1. Screening and Optimizing the Medium Compositions by OFAT

The medium compositions not only influence the production of the target compound, but also influence the economics of the fermentation process. Therefore, the individual effects of different medium components (including the nitrogen source, carbon source, and inorganic salt) on 4′-N-demethyl-vicenistatin production were investigated sequentially based on the AM3 medium.

To achieve the highest 4′-N-demethyl-vicenistatin production, it is critical to maintain a balance between cell growth and secondary metabolite production, which is affected by growth-limiting nutrients such as carbon and nitrogen sources [17]. Consequently, strategies employing readily metabolized carbon sources for cell growth and sustained-release carbon sources for metabolite synthesis are commonly employed in the fermentation of secondary metabolites [18–20]. In addition, it is reported that the secondary metabolite secretion in *Streptomyces* is typically stimulated by sustained-release carbon sources, such as soluble starch, cassava starch, and dextrins [21]. In line with these findings, the effect of different readily metabolized carbon sources on the production of 4′-N-demethyl-vicenistatin

was first studied and the results showed that glycerol produced the highest concentration, followed by glucose and mannose (Figure 1A). Additionally, the results of concentration screening showed that the maximum 4′-N-demethyl-vicenistatin production was achieved at 15 g/L glycerol (Figure 1B). In terms of sustained-release carbon sources, the highest 4′-N-demethyl-vicenistatin production of 0.067 g/L was obtained when cassava starch was used, which showed a significant increase compared to the control (soluble starch), consistent with the previous studies of *Streptomyces* sp. fermentation for neomycin sulfate [22] and epsilon-poly-l-lysine production [23]. In contrast, lactose has a negative effect on 4′-N-demethyl-vicenistatin production (Figure 1C). The screening of the optimum cassava starch concentration showed the maximum 4′-N-demethyl-vicenistatin production was observed at 8 g/L (Figure 1D). The inhibitory effect was observed at a higher carbon source concentration (Figure 1B,D), which could be due to the increased osmotic pressure. Moreover, the osmotic pressure of the fermentation medium in marine microorganisms can be regulated by the concentration of seawater salt, thereby influencing the secretion of secondary metabolites [24]. The optimal salt concentration for 4′-N-demethyl-vicenistatin production was determined to be 30 g/L (Figure 1K), which aligns with the seawater salt concentration in the AM3 medium.

The nitrogen source is an essential element for the microbial synthesis of cellular metabolites, nucleic acids, and proteins. Thus, fermentation media consisting of multiple nitrogen sources to meet microbial nutrient requirements are commonly used [25]. Soybean meal, as an organic nitrogen source, contains numerous nutritious elements, such as a carbohydrate, protein, fatty acid, and trace element contents (potassium, magnesium, sodium, iron, etc.) [26], which has been reported for clavulanic acid production by *Streptomyces clavuligeru* [27] and valinomycin production by *Streptomyces* sp. ZJUT-IFE-354 [28]. Similarly, soybean meal was proved to be an efficient nitrogen source among the selected five nitrogen sources for 4′-N-demethyl-vicenistatin production by *S. parvus* SCSIO Mla-L010/Δ*vicG*. It resulted in a remarkable 66% increase compared to the control (1.5% bacterial peptone and 0.5% soybean meal) (Figure 1E). Surprisingly, it was found that a lower concentration of soybean meal at 5 g/L yielded the highest production of 4′-N-demethyl-vicenistatin (Figure 1F).

Minerals, which act as the cofactors for biosynthetic enzymes to catalyze the necessary reactions, are another necessary component to enhance secondary metabolite secretion in a fermentation medium [29]. For example, $Mo^{6+}$ is normally involved in cell metabolism as the cofactor of reductases, oxidases, and dehydrogenases, and can promote cell proliferation [30] and mupirocin production [31]. $Co^{2+}$ is able to significantly increase the activity of methylmalonyl-CoA mutase and methylmalonyl-CoA transcaboxylase in *S. erythromycin*, as well as fluxes in the glucose metabolism pathway [32]. $Mg^{2+}$ and $Fe^{2+}$ were reported to paly vital functions for lipostatin [33], antimicrobial compounds [34], and staurosporine [35] production. Thus, the effects of $Na_2MoO_4·2H_2O$, $MgSO_4·7H_2O$, $FeSO_4·7H_2O$, and $CoSO_4·7H_2O$ on 4′-N-demethyl-vicenistatin production were evaluated in this study. As shown in Figure 1G, the results demonstrate that only $FeSO_4·7H_2O$ exhibited significant positive effects, while $CoSO_4·7H_2O$ had a detrimental impact on the production of 4′-N-demethyl-vicenistatin among the selected inorganic salts. Moreover, the optimum concentration of $FeSO_4·7H_2O$ was determined to be 40 mg/L (Figure 1H).

Finally, the impact of various inorganic nitrogen sources on the production of 4′-N-demethyl-vicenistatin was investigated. The results demonstrate that ammonium citrate exhibited the highest production (Figure 1I), potentially due to its provision of both ammonium ions and citric acid, which promote microbial growth and substrate utilization [36]. Subsequent optimization revealed that an ammonium citrate concentration of 6 g/L was optimal (Figure 1J).

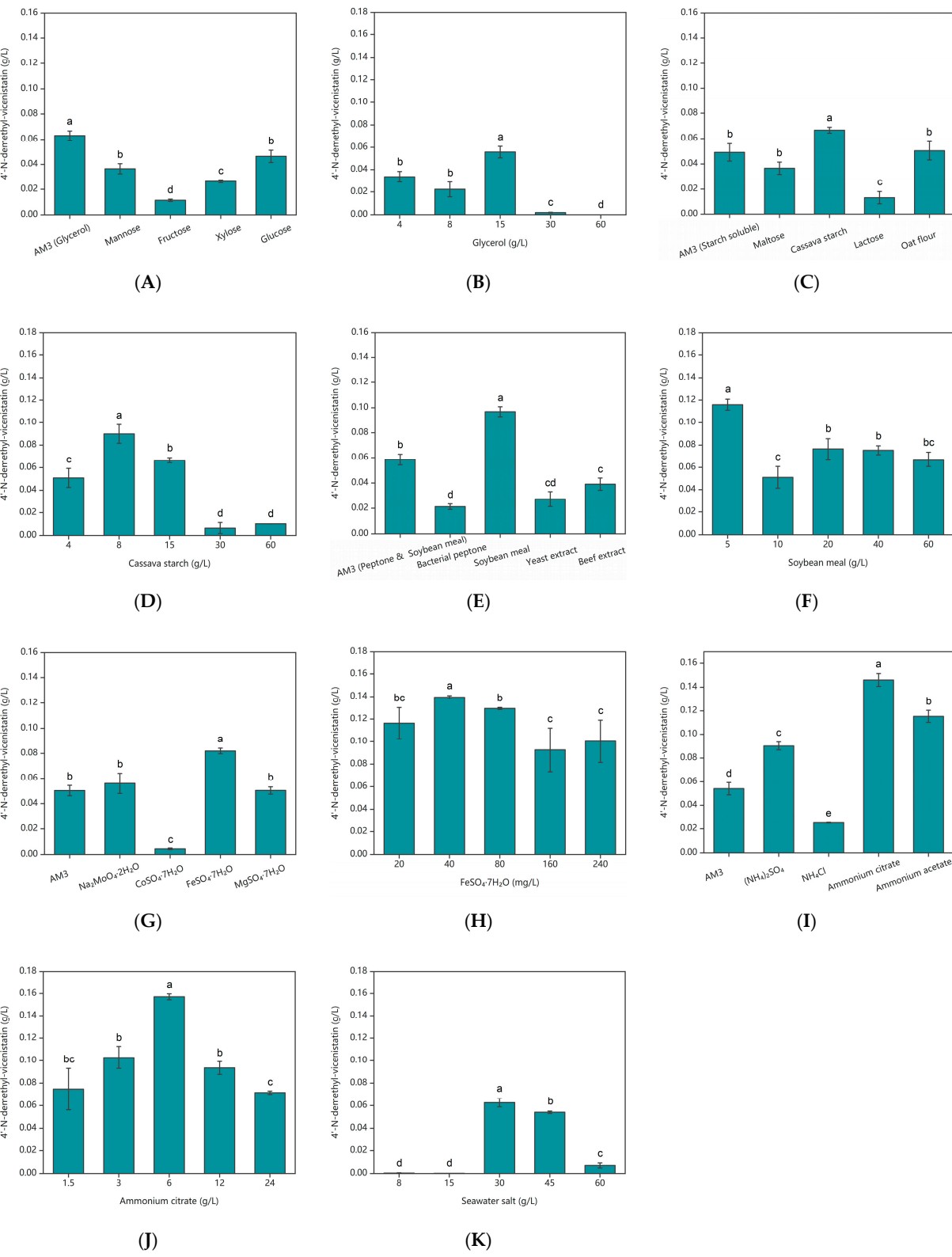

**Figure 1.** Optimization of 4′-N-demethyl-vicenistatin production by *S. parvus* SCSIO Mla-L010/Δ*vicG* using the OFAT approach. (**A**) Effect of readily metabolized carbons, (**B**) Effect of various sustained-release carbons, (**C**) Effect of organic nitrogens, (**D**) Effect of inorganic salts, (**E**) Effect of inorganic nitrogens, (**F**) Effect of glycerol concentration, (**G**) Effect of cassava starch concentration, (**H**) Effect of soybean meal concentration, (**I**) Effect of FeSO₄·7H₂O concentration, (**J**) Effect of ammonium citrate concentration, and (**K**) Effect of seawater salt. The different lowercase letters on the bars indicate statistically significant differences ($p < 0.05$).

### 2.2. Searching for the Most Significant Medium Components by Plackett–Burman Design (PBD)

After conducting a thorough screening and optimization of the medium compositions using the OFAT approach, PBD was employed to identify the key medium components significantly influencing the production of 4′-N-demethyl-vicenistatin. Three concentration levels (−1, 0, and 1) for six factors (cassava starch, glycerol, soybean meal, ammonium citrate, FeSO$_4$·7H$_2$O, and seawater salt) were used in this study, which produced a total of fifteen experimental runs (Table 1). Based on these data, the first-order model equation for the prediction of 4′-N-demethyl-vicenistatin production was obtained and is given below as follows:

$$Y = 0.09062 + 0.02675\ X_1 + 0.02587\ X_2 + 0.00146\ X_3 + 0.00603\ X_4 - 0.00664\ X_5 - 0.01598\ X_6 - 0.0001\ C_T P_T, \tag{1}$$

where $Y$ is the predicted 4′-N-demethyl-vicenistatin concentration; and $X_1$, $X_2$, $X_3$, $X_4$, $X_5$, and $X_6$ are the concentrations of cassava starch, glycerol, soybean meal, ammonium citrate, FeSO$_4$·7H$_2$O, and seawater salt, respectively.

**Table 1.** Plackett–Burman experimental design, response values, and ANOVA.

| Run | $X_1$ Cassava Starch (g/L) | $X_2$ Glycerol (g/L) | $X_3$ Soybean Meal (g/L) | $X_4$ Ammonium Citrate (g/L) | $X_5$ FeSO$_4$·7H$_2$O (mg/L) | $X_6$ Seawater Salt (g/L) | $Y$ 4′-N-Demethyl-Vicenistatin (g/L) |
|---|---|---|---|---|---|---|---|
| 1 | 6(−1) | 10(−1) | 2.5(−1) | 7.5(+1) | 60(+1) | 36(+1) | 0.1070 ± 0.0039 |
| 2 | 10(+1) | 20(+1) | 7.5(+1) | 4.5(−1) | 60(+1) | 36(+1) | 0.1061 ± 0.0684 |
| 3 | 10(+1) | 10(−1) | 7.5(+1) | 4.5(−1) | 30(−1) | 24(−1) | 0.1099 ± 0.0214 |
| 4 | 10(+1) | 10(−1) | 7.5(+1) | 7.5(+1) | 30(−1) | 36(+1) | 0.1042 ± 0.0203 |
| 5 | 6(−1) | 20(+1) | 7.5(+1) | 4.5(−1) | 60(+1) | 24(−1) | 0.1173 ± 0.0575 |
| 6 | 10(+1) | 20(+1) | 2.5(−1) | 7.5(+1) | 30(−1) | 24(−1) | 0.1609 ± 0.0195 |
| 7 | 8(0) | 15(0) | 5(0) | 6(0) | 40(0) | 30(0) | 0.0832 ± 0.0211 |
| 8 | 6(−1) | 10(−1) | 2.5(−1) | 4.5(−1) | 30(−1) | 24(−1) | 0.0606 ± 0.0066 |
| 9 | 10(+1) | 10(−1) | 2.5(−1) | 4.5(−1) | 60(+1) | 36(+1) | 0.0564 ± 0.0075 |
| 10 | 8(0) | 15(0) | 5(0) | 6(0) | 40(0) | 30(0) | 0.0922 ± 0.0162 |
| 11 | 6(−1) | 20(+1) | 2.5(−1) | 4.5(−1) | 30(−1) | 36(+1) | 0.0571 ± 0.0060 |
| 12 | 6(−1) | 10(−1) | 7.5(+1) | 7.5(+1) | 60(+1) | 24(−1) | 0.0241 ± 0.0008 |
| 13 | 6(−1) | 20(+1) | 7.5(+1) | 7.5(+1) | 30(−1) | 36(+1) | 0.0908 ± 0.0251 |
| 14 | 8(0) | 15(0) | 5(0) | 6(0) | 40(0) | 30(0) | 0.0960 ± 0.0213 |
| 15 | 10(+1) | 20(+1) | 2.5(−1) | 7.5(+1) | 60(+1) | 24(−1) | 0.1667 ± 0.0139 |
| | | | | | | | Model |
| Adj SS | 0.008588 | 0.008028 | 0.000026 | 0.000436 | 0.000529 | 0.003064 | 0.020671 |
| Adj MS | 0.008588 | 0.008028 | 0.000026 | 0.000436 | 0.000529 | 0.003064 | 0.002953 |
| F-Value | 23.49 | 21.96 | 0.07 | 1.19 | 1.45 | 8.38 | 8.08 |
| *p*-Value | 0.002 | 0.002 | 0.799 | 0.311 | 0.268 | 0.023 | 0.007 |

The analysis of variance (ANOVA) indicated that the model was statistically significant (*p* = 0.007) (Table 1). Among the six factors, cassava starch and glycerol were extremely significant (*p* < 0.01), seawater salt was significant (*p* < 0.05), and soybean meal, ammonium citrate, and FeSO$_4$·7H$_2$O were not significant (*p* > 0.05).

The steepest ascent design (SAD) was used to search for the highest response rates by increasing the concentrations of cassava starch and glycerol (positive coefficient), while decreasing the concentration of seawater salt (negative coefficient) (Table 2). As a result, the highest 4′-N-demethyl-vicenistatin concentration was obtained at 11 g/L cassava starch, 18 g/L glycerol, and 27.5 g/L seawater salt. To sum up, cassava starch, glycerol, and seawater salt were identified as the most significant medium components affecting 4′-N-demethyl-vicenistatin production, and this point (11 g/L cassava starch, 18 g/L glycerol, and 28 g/L seawater salt) was selected for further optimization.

**Table 2.** Steepest ascent experiment design and response values.

| Run | Cassava Starch | Glycerol | Seawater Salt | 4′-N-Demethyl-Vicenistatin |
|---|---|---|---|---|
| | (g/L) | (g/L) | (g/L) | (g/L) |
| Step size | +3 | +3 | −2.5 | |
| 1 | 8 | 15 | 30 | 0.1070 ± 0.0037 |
| 2 | 11 | 18 | 27.5 | 0.1399 ± 0.0036 |
| 3 | 14 | 21 | 25 | 0.0159 ± 0.0007 |
| 4 | 17 | 24 | 22.5 | 0.0022 ± 0.0003 |
| 5 | 20 | 27 | 20 | 0.0013 ± 0.0007 |

*2.3. Modeling and Optimization of Medium Compositions by Response Surface Methodology (RSM)*

Based on the results of PBD and SAD, the optimal proportion of the medium components were further optimized by RSM using three-factor, five-level central composite design (CCD) (Table 3). Based on the data of 20 experimental runs, the multivariate non-linear regression equation associated with the production prediction model is presented as follows:

$$Y = 0.13993 - 0.00748X_1 + 0.00811X_2 + 0.00630X_3 - 0.02123X_1^2 - 0.05481X_2^2 + 0.00848X_3^2 - 0.02940X_1X_2 + 0.01521X_1X_3 - 0.02487X_2X_3, \tag{2}$$

where $Y$ is the predicted 4′-N-demethyl-vicenistatin concentration; and $X_1$, $X_2$, and $X_3$ are concentrations of cassava starch, glycerol, and seawater salt, respectively.

**Table 3.** Central composite design with experimental data and predicted data by RSM.

| Run | $X_1$ | $X_2$ | $X_3$ | $Y$ |
|---|---|---|---|---|
| | Cassava Starch | Glycerol | Seawater Salt | 4′-N-Demethyl-Vicenistatin |
| | (g/L) | (g/L) | (g/L) | (g/L) |
| 1 | 11(0) | 9(−1.5) | 28(0) | 0.0157 ± 0.0049 |
| 2 | 5(−1) | 24(+1) | 34(+1) | 0.0896 ± 0.0414 |
| 3 | 5(−1) | 12(−1) | 22(−1) | 0.0266 ± 0.0098 |
| 4 | 5(−1) | 12(−1) | 34(+1) | 0.0539 ± 0.0100 |
| 5 | 5(−1) | 24(+1) | 22(−1) | 0.1566 ± 0.0275 |
| 6 | 17(+1) | 12(−1) | 22(−1) | 0.0330 ± 0.0075 |
| 7 | 11(0) | 18(0) | 28(0) | 0.1323 ± 0.0274 |
| 8 | 11(0) | 18(0) | 28(0) | 0.1424 ± 0.0380 |
| 9 | 11(0) | 18(0) | 19(−1.5) | 0.1476 ± 0.1049 |
| 10 | 11(0) | 18(0) | 28(0) | 0.1443 ± 0.0048 |
| 11 | 11(0) | 18(0) | 37(+1.5) | 0.1719 ± 0.0388 |
| 12 | 2(−1.5) | 18(0) | 28(0) | 0.0982 ± 0.0011 |
| 13 | 17(+1) | 24(+1) | 34(+1) | 0.0392 ± 0.0028 |
| 14 | 11(0) | 18(0) | 28(0) | 0.1453 ± 0.0061 |
| 15 | 17(+1) | 12(−1) | 34(+1) | 0.1263 ± 0.0066 |
| 16 | 17(+1) | 24(+1) | 22(−1) | 0.0505 ± 0.0596 |
| 17 | 11(0) | 27(+1.5) | 28(0) | 0.0191 ± 0.0008 |
| 18 | 20(+1.5) | 18(0) | 28(0) | 0.0877 ± 0.0090 |
| 19 | 11(0) | 18(0) | 28(0) | 0.1372 ± 0.0037 |
| 20 | 11(0) | 18(0) | 28(0) | 0.1370 ± 0.0179 |

The adequacy and fitness of this model were statistically analyzed using ANOVA, as shown in Table 4. In general, it has a low *p*-value ($0.183 \times 10^{-7}$) and high F-value (93.65). In addition, the *p*-value and F-value of the "lack-of-fit" were 0.083 and 3.85, respectively, demonstrating that this model was statistically significant and could explain the responses accurately. The high $R^2$ of 0.9883 and Adj $R^2$ of 0.9777 indicated that it was reliable to use

this model to predict the 4′-N-demethyl-vicenistatin production. Moreover, the *p*-values for the linear, squared, and interaction terms were less than 0.05, indicating that they all had a significant influence on 4′-N-demethyl-vicenistatin production.

**Table 4.** ANOVA for the RSM model.

| Source | DF | Adj SS | Adj MS | F-Value | *p*-Value |
|---|---|---|---|---|---|
| Model | 9 | 0.052301 | 0.005811 | 93.65 | $0.183 \times 10^{-7}$ |
| $X_1$ | 1 | 0.000699 | 0.000699 | 11.26 | 0.007 |
| $X_2$ | 1 | 0.000822 | 0.000822 | 13.25 | 0.005 |
| $X_3$ | 1 | 0.000496 | 0.000496 | 7.99 | 0.018 |
| $X_{12}$ | 1 | 0.004644 | 0.004644 | 74.84 | $0.590 \times 10^{-5}$ |
| $X_{22}$ | 1 | 0.030963 | 0.030963 | 498.96 | $0.727 \times 10^{-9}$ |
| $X_{32}$ | 1 | 0.000741 | 0.000741 | 11.94 | 0.006 |
| $X_1 \times X_2$ | 1 | 0.006916 | 0.006916 | 111.45 | $0.965 \times 10^{-6}$ |
| $X_1 \times X_3$ | 1 | 0.001852 | 0.001852 | 29.84 | $0.276 \times 10^{-3}$ |
| $X_2 \times X_3$ | 1 | 0.004950 | 0.004950 | 79.76 | $0.443 \times 10^{-5}$ |
| Error | 10 | 0.000621 | 0.000062 | | |
| Lack-of-Fit | 5 | 0.000493 | 0.000099 | 3.85 | 0.083 |
| Pure Error | 5 | 0.000128 | 0.000026 | | |
| Total | 19 | 0.052922 | | | |
| $R^2 = 0.9883$ | | Adj $R^2 = 0.9777$ | | RMSE = 0.0056 | |

Based on the RSM analysis, 3D surface plots were produced to visualize the interactive effects among the experimental variables (Figure 2). As shown in Figure 2A, 4′-N-demethyl-vicenistatin production increased with an increasing concentration of cassava starch and glycerol until the central level (0), when the concentration of seawater salt was fixed. The lowest concentration values were observed at high or low cassava starch and glycerol combinations. Figure 2B shows the interaction of cassava starch and seawater salt on 4′-N-demethyl-vicenistatin production, when the concentration of glycerol was fixed. The highest concentration was observed at a high concentration of seawater salt and a moderate concentration of cassava starch, with the lowest concentration being observed on the high cassava starch and low seawater salt combinations. It can be seen from Figure 2C that a high concentration of 4′-N-demethyl-vicenistatin was obtained at a central-level glycerol concentration and a high/low-level seawater salt concentration. In summary, the order of significance of these three variables on 4′-N-demethyl-vicenistatin production was as follows: glycerol > cassava starch > seawater salt (Table 4). The model predicted that the highest 4′-N-demethyl-vicenistatin concentration (0.1848 g/L) would be obtained when the medium contained 4 g/L cassava starch, 22 g/L glycerol, and 19 g/L seawater salt.

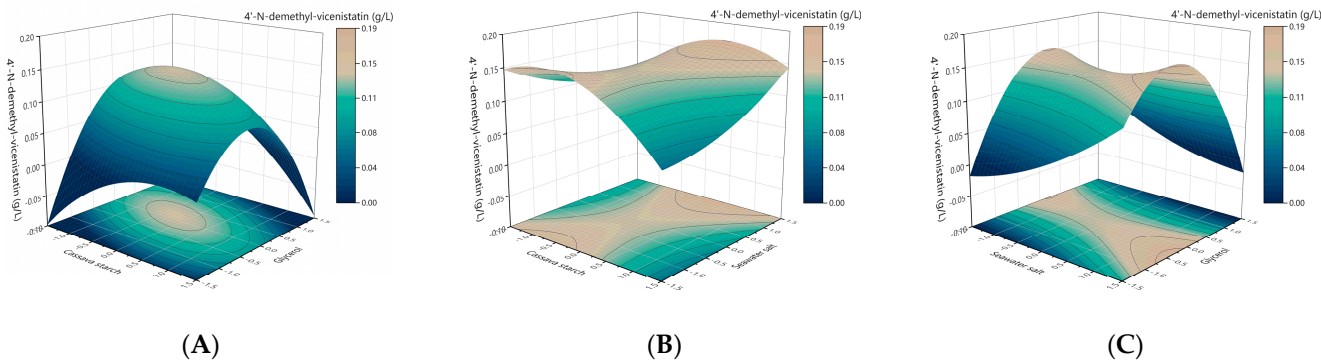

**(A)**  **(B)**  **(C)**

**Figure 2.** Effects of variables on 4′-N-demethyl-vicenistatin production. (**A**) Cassava starch versus glycerol; (**B**) Cassava starch versus seawater salt; (**C**) Seawater salt versus glycerol.

### 2.4. Modeling and Optimization of Medium Compositions by Artificial-Neural-Network-Genetic-Algorithm (ANN-GA)

The backpropagation (BP) learning algorithm is a commonly employed method for training artificial neural networks (ANNs) and is frequently utilized as a statistical data modeling tool in the optimization process of fermentation [16,37]. In this study, the widely adopted technique of k-fold cross-validation was implemented to address the small sample sizes during the training process of ANNs (Figure 3A) [38]. Additionally, the GA model is commonly employed as an optimization tool for identifying optimal conditions, utilizing the trained ANN as the fitness function [39,40].

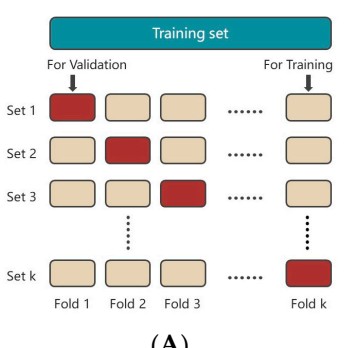 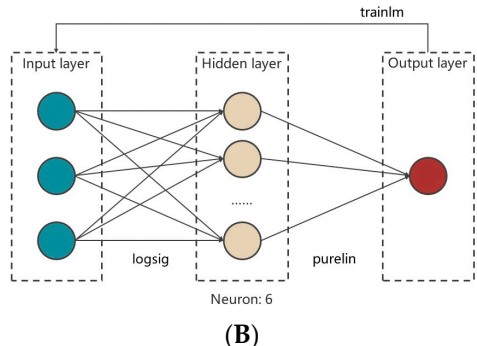

**(A)** **(B)**

**Figure 3.** Schematic diagram of the main process of k-fold cross-validation (**A**) and the architecture of the optimum ANN model (**B**).

Typically, prior to training an ANN model, the optimal number of neurons in the hidden layer, training algorithm, and activation function are first determined based on minimizing the MSE [16,41,42]. In terms of neurons in the hidden layer, as shown in Table 5, the MSE value fluctuated in the range of 0.2588 to 0.3331 when increasing the number of neurons from 3 to 12. The optimal architecture of six neurons was selected due to its lowest MSE value (0.2588). Therefore, 3-6-1 topology was finally chosen to estimate 4′-N-demethyl-vicenistatin production. The selection of appropriate transfer functions and backpropagation training algorithm is an essential step in the design of ANN. To determine the optimum combination of transfer functions for hidden and output layers, the comparison of six different combinations was performed, with the training algorithm chosen as 'traingd.' The results showed that the combination of logsig and purelin yielded the lowest MSE value of 0.2177 (Table 5). Then, 11 different backpropagation training algorithms (as shown in Table 5) were tested to search for an applicable training algorithm for the accurate prediction of 4′-N-demethyl-vicenistatin production. The optimum training algorithm of trainlm was selected because it obtained the minimum MSE value of 0.2453. The finalized ANN architecture (3-6-1) containing transfer functions and the backpropagation training algorithm for the prediction of 4′-N-demethyl-vicenistatin production is summarized in Figure 3B.

To construct the ANN model successfully, a complete set of data from CCD was divided into a training set (14), testing set (3), and validation set (3). Figure 4A clearly shows the performance evaluation results of the ANN model during the training, testing, and validation. The developed ANN achieved the best validation performance at three epochs, with an MSE value of 0.0139. The training was stopped at five epochs, since the validation error did not decrease continuously for five epochs. Figure 4B displays the distribution of the fitting error of the training data. The fitting errors on most of the training data were quite close to zero and spread over a reasonable range. Figure 4C shows that good R values were obtained for training (0.999), validation (0.995), test (0.999), and all combined (0.996), respectively, suggesting an excellent correlation between the actual and predicted values.

**Table 5.** Optimization of the number of neurons, transfer functions, and backpropagation training algorithm for the ANN model.

| Parameters | | MSE |
|---|---|---|
| Number of neurons | 3 | 0.3331 |
| | 4 | 0.3181 |
| | 5 | 0.3138 |
| | 6 | 0.2588 |
| | 7 | 0.2733 |
| | 8 | 0.2833 |
| | 9 | 0.2789 |
| | 10 | 0.2812 |
| | 11 | 0.3034 |
| | 12 | 0.2868 |
| Transfer functions | logsig + purelin | 0.2177 |
| | tansig + purelin | 0.2539 |
| | logsig + tansig | 0.2202 |
| | tansig + logsig | 0.3618 |
| | tansig + tansig | 0.2703 |
| | logsig + logsig | 0.3234 |
| Backpropagation training algorithm | trainbr | 0.4448 |
| | traincgb | 0.3799 |
| | traincgf | 0.3392 |
| | traincgp | 0.3746 |
| | traingd | 0.4474 |
| Backpropagation training algorithm | traingda | 0.4341 |
| | traingdm | 0.5184 |
| | traingdx | 0.5052 |
| | trainlm | 0.2453 |
| | trainrp | 0.3207 |
| | trainscg | 0.3811 |

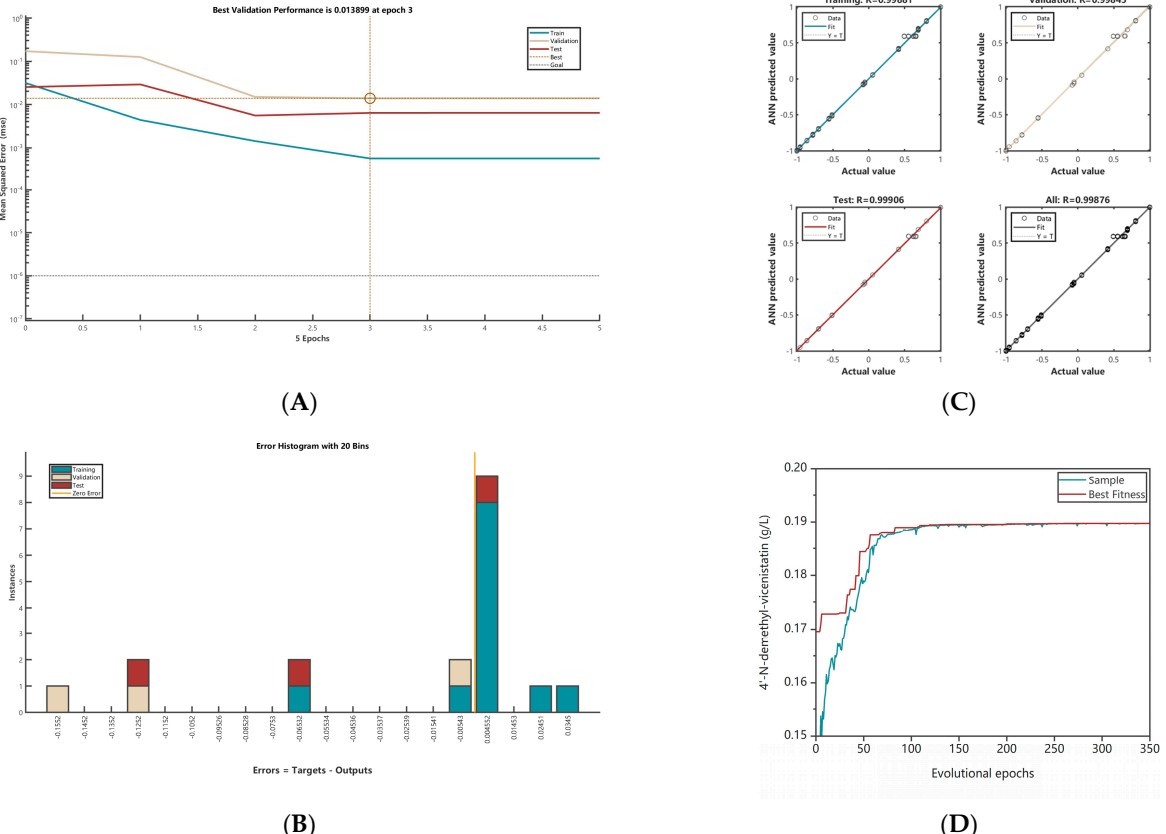

(**A**)

(**B**)

(**C**)

(**D**)

**Figure 4.** The results of performance plot (**A**), error histogram (**B**), regression plot (**C**), and GA optimization process (**D**) for the constructed ANN-GA model.

Finally, GA was performed to give an optimized solution in a region of the parameter space with the trained ANN model as the fitness function [43]. As shown in Figure 4D, the maximal 4'-N-demethyl-vicenistatin production of 0.1885 g/L was achieved at the following medium compositions: 12 g/L cassava starch, 17 g/L glycerol, and 34 g/L seawater salt.

## 2.5. Comparison and Assessment of RSM and ANN-GA Models

The comparison and evaluation of ANN-GA and RSM were conducted based on the models' *error*, $R^2$, MSE, RMSE, and APD values, as well as the radar map and parity plot. First, in comparison to the RSM model, the ANN-GA model exhibits superior predictive and extrapolative capabilities, as evidenced by its lower error value (ANN-GA: 1.90% vs. RSM: 11.40%), lower APD value (ANN-GA: 5.88 vs. RSM: 10.58), and lower RMSE value (ANN-GA: 0.0051 vs. RSM: 0.0056) (Table 6). In addition, the statistical evaluation of the model's predictive values for 4'-N-demethyl-vicenistatin production is visually depicted using a radar map (Figure 5A) and parity plot (Figure 5B), based on the predicated data from two models utilizing 20 sets of experimental operating conditions, as listed in the CCD design. The results depicted in Figure 5 demonstrate a high level of coincidence and accuracy between both of the models with the experimental data. However, the ANN-GA model exhibited a superior predictive performance compared to the RSM model in this study, as evidenced by its higher $R^2$ value (ANN-GA: 0.9962 vs. RSM: 0.9883) and lower MSE value (ANN-GA: $2.62 \times 10^{-5}$ vs. RSM: $3.11 \times 10^{-5}$) (Table 6). In the end, the RSM and ANN-GA models were finally experimentally verified for 4'-N-demethyl-vicenistatin production (Table 6). In regard to the RSM model, 0.1637 g/L 4'-N-demethyl-vicenistatin was reached using the optimized fermentation medium containing 4 g/L cassava starch, 22 g/L glycerol, and 19 g/L seawater salt, which was significantly inferior to the predicted value of 0.1848 g/L. In contrast, the experimental 4'-N-demethyl-vicenistatin concentration was 0.1921 g/L in the ANN-GA-optimized medium consisting of 12 g/L cassava starch, 17 g/L glycerol, and 34 g/L seawater salt, which was comparable to the predicted value of 0.1885 g/L. It should be noted that the 4'-N-demethyl-vicenistatin concentration obtained by using the ANN-GA-optimized medium was 17% and 283% higher than those achieved using the RSM-optimized medium and the initial medium, respectively. Although the RSM model has been widely employed for process parameter optimization, its application has been limited due to its ability to only construct a second-order polynomial regression model. In contrast, the ANN-GA model possesses the capability to predict almost all forms of nonlinearity [44]. Therefore, all of these results indicated that the ANN-GA model had a better optimization and prediction capability than the RSM model.

**Table 6.** Comparison of medium compositions and prediction capability between RSM and ANN-GA.

| Model | Process Parameter | | | Statistical Values | | | | | 4'-N-Demethyl-Vicenistatin (g/L) | |
|---|---|---|---|---|---|---|---|---|---|---|
| | Cassava Starch | Glycerol | Seawater Salt | Error | $R^2$ | MSE | RSME | APD | Experimental Data | Predicted Data |
| | (g/L) | (g/L) | (g/L) | | | | | | | |
| AM3 | 15 | 15 | 30 | - | - | - | - | - | 0.0502 ± 0.0041 | - |
| RSM | 4 | 22 | 19 | 11.4% | 0.9883 | $3.11 \times 10^{-5}$ | 0.0056 | 10.58 | 0.1637 ± 0.0036 | 0.1848 |
| ANN-GA | 12 | 17 | 34 | 1.9% | 0.9962 | $2.62 \times 10^{-5}$ | 0.0051 | 5.88 | 0.1921 ± 0.0052 | 0.1885 |

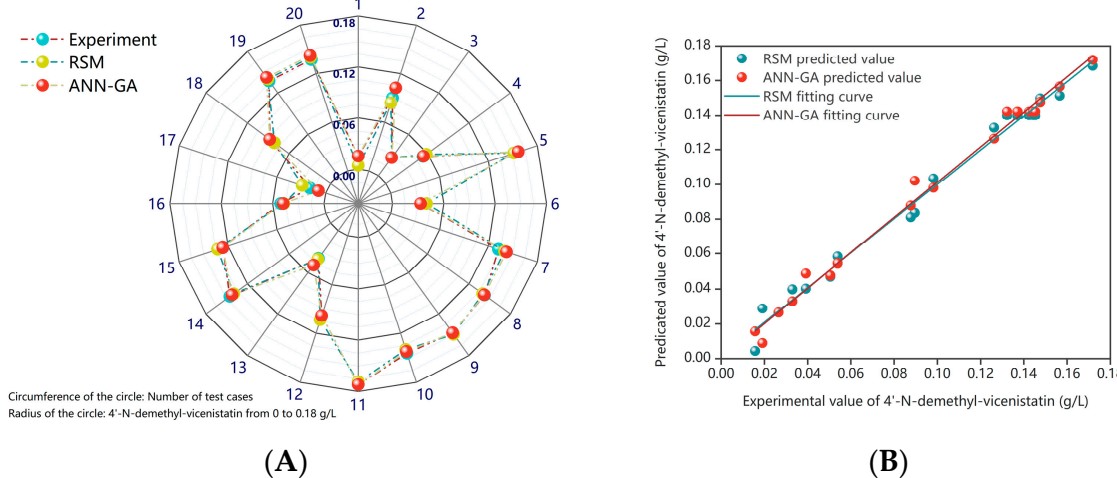

**Figure 5.** The results of radar map (**A**) and parity plot (**B**) were compared between the CCD model and the ANN-GA model.

## 3. Discussion

The optimized medium formulation (12 g/L cassava starch, 17 g/L glycerol, 34 g/L seawater salt, 7.5 g/L soybean meal, 7.5 g/L ammonium citrate, 30 mg/L $FeSO_4 \cdot 7H_2O$, 2 g/L calcium carbonate) achieved through the implementation of OFAT, CCD, and ANN-GA in this study resulted in a remarkable 38-fold increase in yield compared to the previously reported value of approximately 5 mg/L [11]. Moreover, 4′-N-demethyl-vicenistatin exhibited excellent antibacterial activity and displayed promising potential for patenting purposes [11]. Consequently, this enhanced production can serve as a solid foundation for subsequent derivatization experiments and other research and development endeavors.

The focus of this discussion will be on the following two key aspects: the optimization strategy for medium formulation and the selection of research methods.

The first aspect pertains to the optimization of the medium formulation strategy. This study revealed that the composition and proportion of carbon sources (glycerol and cassava starch), in conjunction with seawater salt concentration, exerted a significant influence on secondary metabolite production (4′-N-demethyl-vicenistatin) by *S. parvus* SCSIO Mla-L010/Δ*vicG*. The *p*-values of glycerol, cassava starch, and seawater salt in Table 1 were all found to be less than 0.05, indicating the significant impact of their concentration changes on the production of 4′-N-demethyl-vicenistatin. In comparison to the formula (15 g/L glycerol, 15 g/L soluble starch, 30 g/L seawater salt) utilized in the AM3 medium, the optimized formula (17 g/L glycerol, 12 g/L cassava starch, 34 g/L seawater salt) predicted by the ANN-GA model resulted in an increased yield, from 0.0502 g/L to 0.1921 g/L (Table 6). The composition and concentration of the medium can exert a significant influence on microbial fermentation production [45–47]. This is particularly applicable to microbial secondary metabolic processes, where the composition and concentration of the carbon sources in the medium can significantly impact the fermentation of secondary metabolites, such as pigment fermentation and antibiotic production [48–50]. The choice of carbon source can exert a significant influence on the biosynthesis of specific secondary metabolites, owing to its capacity to inhibit gene expression and repress enzyme activity [51]. For example, the production of cephamycin C, a β-lactam antibiotic synthesized by *Streptomyces clavuligerus*, encounters challenges in the presence of glycerol, due to its suppressive impact on the activity of cephamycin C synthetase and expandase enzymes [52]. Glucose exerts a suppressive effect on *afsR2* mRNA synthesis, encoding a global regulatory protein responsible for facilitating secondary metabolite biosynthesis in *Streptomyces lividans*, thereby resulting in the inhibition of actinorhodin (a polyketide) production [53]. While glycerol enhances cellular growth and internal ATP levels, it hinders the synthesis of spiramycin. Spiramycin, a potent macrolide antibiotic derived from *Streptomyces ambofaciens*, is commonly prescribed

for the treatment of toxoplasmosis. The presence of glucose and glycerol adversely impacts the production of spiramycin [54]. Therefore, in order to exert better control over the concentration and type of carbon source, numerous researchers opt for feeding carbon into the fed-batch fermentation process. For example, the enhanced production of secondary metabolites in *Inonotus obliquus*, a traditional medicinal fungus utilized for cancer and other ailments, by 65% is conducted employing a glucose-fed batch dissolved oxygen (DO) control strategy. This approach entails supplementation with 10 g/L glucose upon reaching a residual sugar concentration of 10 g/L while maintaining the DO level at 50% [55]. The production of alkaline amylase in *B. subtilis* 168 mut-16# strain was significantly enhanced to 591.4 U/mL by optimizing the agitation speed and supplementing with hydrolyzed starch during the 10th hour of fermentation [56]. The *Streptomyces graminearus* F3-4 strain was employed in a fed-batch fermentation process, resulting in the production of epsilon-PL reaching a maximum concentration of 13.5 g/L. This remarkable production was achieved by increasing the initial glucose concentration from 50 to 85 g/L. To maintain optimal conditions, the supplementary mixture was manually introduced into the broth when the glucose concentration dropped to 0.5%, ensuring its final concentration reached 1.5% [57].

Additionally, employing a diverse range of carbon sources in the culture medium is a judicious approach. This strategy entails harnessing both readily metabolized carbon sources (monosaccharides) and sustained-release carbon sources (polysaccharides). By capitalizing on *Streptomyces*' capacity to hydrolyze polysaccharides such as amylase, the microorganisms can initially utilize easily accessible carbon sources for growth and subsequently synthesize their own hydrolases to metabolize and utilize complex polysaccharides. As presented in Table 6, the RSM model formulation incorporates a concentration of 22 g/L of readily metabolized carbon sources (glycerol) and 4 g/L of sustained-release carbon sources (cassava starch), whereas the ANN-GA model formulation comprises 12 g/L of readily metabolized carbon sources (glycerol) and 17 g/L of sustained-release carbon sources (cassava starch). By optimizing the concentration and composition of carbon sources, the production of the ANN-GA model exhibited a significant increase from 0.1637 g/L (RSM model) to 0.1921 g/L (ANN-GA model). In the absence of feeding extra carbons, the microorganisms autonomously regulate the concentration and composition of sugars in the culture medium, thereby circumventing some of the potential impact on the secondary metabolism induced by carbon sources [58–60]. For example, the production of cold-adapted beta amylase from *Streptomyces* was enhanced by Cotârlet et al. through the utilization of a medium formulation comprising glycerol and starch, with an optimized ratio [61,62]. Smaoui [63], Al-Ansari [64], and Ni [65] et al. opted for a medium formulation comprising glucose and starch and meticulously optimized the proportions of these constituents to enhance the yields. However, different *Streptomyces* species may exhibit diverse metabolic mechanisms for utilizing multiple carbon sources. For instance, *Streptomyces albulus* M-Z18 demonstrates the ability to efficiently utilize both glucose and glycerol in a manner unaffected by the presence of glucose. Zeng et al. successfully enhanced ε-poly-L-lysine productivity from *S.albulus* M-Z18 by supplementing the growth medium with a combination of glucose and glycerol [66]. Additionally, glycerol exerts a beneficial effect on the biosynthesis of antibacterial compounds in certain strains of Streptomyces [67,68]. In conclusion, there exists a scientific foundation for *S. parvus* SCSIO Mla-L010/Δ*vicG* to achieve enhanced production through the utilization of a medium with an optimized ratio of glycerol to starch. However, further investigations into the carbon metabolism and polyketide synthase (PKS) mechanisms of *S. parvus* SCSIO Mla-L010/Δ*vicG* are imperative for production improvement.

The strategy used to enhance the yield by manipulating the seawater salt concentration is predicated on the principle that microorganisms maintain equilibrium between internal and external osmotic pressures through endogenous synthesis, environmental absorption, and other mechanisms for counteracting external osmotic pressure [69,70]. In particular, marine microorganisms, such as *Streptomyces*, isolated from the sea often exhibit significant alterations in their secondary metabolism due to variations in seawater salt concentration.

For instance, Sanjivkumar, Selvaraj, and Manivasagan et al., respectively, optimized the sea salt proportion in the medium to enhance the production of chitinase in *Streptomyces olivaceus* MSU3 [71], antibiotics in *Streptomyces* sp. CMSTAAHAL-3 [72], and α-amylase in *Streptomyces* sp. MBRC-82 [73]. In this study, *S. parvus* SCSIO Mla-L010/Δ*vicG* exhibited a remarkable tolerance to high osmotic pressure (twice that of the NaCl concentration in seawater) and demonstrated the production of 4′-N-demethyl-vicenistatin. However, the synthesis pathway of 4′-N-demethyl-vicenistatin was found to be obstructed under high osmotic pressure. Surprisingly, through harnessing the synergistic interaction between the sea salt and the carbon source in the medium, augmenting the sea salt concentration by 4 g/L significantly amplified its production.

The second aspect pertains to the selection of research methods. In this study, the ANN-GA model demonstrated a superior performance over the RSM model in optimizing the production of 4′-N-demethyl-vicenistatin by *S. parvus* SCSIO Mla-L010/Δ*vicG*, both in terms of example verification and overall model performance. It has been widely acknowledged that researchers in the field of microbial fermentation optimization commonly employ conventional statistical methods, including the OFAT approach [74], orthogonal experiments [75], PBD [76–78], and RSM [14,79,80], for optimization purposes. For instance, the RSM, whether it is the Box–Behnken design (BBD) or CCD, is generally applicable when there are no more than four experimental factors. Consequently, by designing a lesser number of experimental groups, one can effectively explore the influence of factor interactions on the response value while employing multivariate quadratic polynomials to represent the constructed model [81]. However, the limitations of simple binary equations for accurately elucidating the intricate mechanisms involved in microbial fermentation necessitate the utilization of ANN, which possess the capability to construct more sophisticated models and generally outperform RSM when confronted with such challenges [39,82–85].

The ANN is a computational model composed of interconnected nodes, also known as neurons, wherein each node represents an activation function that determines a specific output. The interconnections between the nodes are represented by weighted values called weights, which serve as the memory component of the ANN. Consequently, the network's output is contingent upon these connections, weight values, and activation functions [86]. The ANN model has the following advantages: Firstly, it possesses the capability of autonomous learning. The autonomous learning function is particularly crucial for prediction. It is anticipated that ANN will offer extensive economic forecasts, market predictions, and benefit projections, thereby rendering its application prospects highly promising. Secondly, ANN exhibits an associative storage function, which can be achieved through its own feedback networks. Thirdly, it demonstrates high-speed computational abilities to rapidly identify optimal solutions. Solving complex problems often necessitates extensive calculations; however, by employing a specific feedback artificial neural network, optimal solutions can be swiftly obtained [87]. Thanks to the development of AI technologies, the rise of ChatGPT, and the advent of Python, non-computer domain researchers now have the opportunity to use ANN to solve problems more conveniently. Furthermore, artificial neural networks (ANN) exhibit exceptional performance in some specific domains, such as the modeling of drinking water quality [88], thermal analysis [89], the design of model predictive control system [90], pattern recognition [91], and photovoltaic fault detection and diagnosis [92], among others. This also encompasses applications in biology [93–95].

The application of ANN in biological fields, particularly when combined with RSM, has demonstrated remarkable outcomes, even with a limited number of samples. However, caution should be exercised due to the potential risk of inaccurate fitting arising from the constrained sample size. Consequently, to enhance the generalization capability of ANN models, some researchers opt for incorporating k-fold cross-validation during the training process [96,97]. Additionally, the training process of an ANN model necessitates the meticulous configuration of numerous parameters. Inadequate parameter settings can lead to either overfitting or underfitting, thereby compromising the model's predictive capacity. Relevant experiments for determining these parameters are typically devised

by researchers either employing OFAT designs or establishing parameter values based on their own expertise [98,99]. In recent years, researchers have increasingly employed GA, particle swarm optimization (PSO), artificial bee colony (ABC), and the backtracking search algorithm (BSA), in conjunction with ANN models, to enhance the efficiency of optimizing training parameters for ANN models, enabling them to address progressively intricate problems more effectively [100].

In this study, after employing k-fold cross-validation and OFAT to optimize the training parameters, an ANN model (3-6-1) was ultimately utilized to investigate the impact of variations in cassava starch, glycerol, and seawater salt concentrations on 4′-N-demethyl-vicenistatin production. Based on the knowledge obtained here, it can be understood that this ANN computational model encompasses 21 functions with diverse weights, facilitating the more precise prediction of outcomes compared to the ternary quadratic polynomial of the RSM model. Therefore, as illustrated in Table 6, the ANN model exhibits a superior performance compared to the RSM model, as evidenced by its lower error value (ANN-GA: 1.90% vs. RSM: 11.40%), reduced APD value (ANN-GA: 5.88 vs. RSM: 10.58), and decreased RMSE value (ANN-GA: 0.0051 vs. RSM: 0.0056). Furthermore, as depicted in Figure 5, the experimental results align more closely with those expected from the ANN-GA model during validation.

The utilization of RSM-ANN in optimizing the medium formulation may consequently yield superior outcomes compared to those achieved solely through RSM.

## 4. Materials and Methods

### 4.1. Bacterial Materials, Culture Medium, and Fermentation Conditions

*S. parvus* SCSIO Mla-L010/Δ*vicG*, a N-methyltransferase knock-out mutant of *S. parvus* SCSIO Mla-L010, which can secrete 4′-N-demethyl-vicenistatin, was used in this study [11]. It was maintained on an agar plate (1% soybean meal, 1% crushed soybean, 2% mannitol, 2.5% agar) and stored at 4 °C.

*S. parvus* SCSIO Mla-L010/Δ*vicG* was first grown on an agar plate at 30 °C for sporulation and the spore was inoculated into 250 mL a shake flask containing 50 mL seed medium (3% TSB medium, 0.5% yeast extract, 10% sucrose) at 200 rpm and 28 °C for 36 h. Subsequently, 5 mL of the seed medium was inoculated into a new 250 mL shake flask containing 50 mL fermentation medium at 200 rpm and 28 °C for 7 days. The AM3 medium, which contained 1.5% glycerol, 1.5% soluble starch, 1.5% bacterial peptone, 0.5% soybean meal, 3% seawater salt, and 0.2% calcium carbonate, was selected as the initial medium [101].

### 4.2. Medium Optimization for Efficient 4′-N-Demethyl-Vicenistatin Production

4.2.1. Screening and Optimizing the Medium Compositions by One-Factor-at-a-Time (OFAT)

OFAT was applied to screen the optimum medium compositions for 4′-N-demethyl-vicenistatin production, with AM3 medium as the initial medium. Glucose, mannose, fructose, xylose, and glycerol (at 15 g/L) were investigated as the readily metabolized carbon sources. Soluble starch, lactose, maltose, cassava starch, and oat flour (at 15 g/L) were used as the sustained-release carbon sources. Bacterial peptone, soybean meal, yeast extract, beef extract, and a mixture of bacterial peptone and soybean meal (at 20 g/L) were selected as the organic nitrogen sources. $NH_4Cl$, $(NH_4)_2SO_4$, ammonium acetate, and ammonium citrate (at 3 g/L) were chosen as the inorganic nitrogen sources. $Na_2MoO_4 \cdot 2H_2O$, $MgSO_4 \cdot 7H_2O$, $FeSO_4 \cdot 7H_2O$, and $CoSO_4 \cdot 7H_2O$ (at 80 mg/L) were investigated as the inorganic salts. After the compositions of the medium were determined, the optimum concentrations of glycerol (4, 8, 15, 30, and 60 g/L), cassava starch (4, 8, 15, 30, and 60 g/L), ammonium citrate (1.5, 3, 6, 12, and 24 g/L), $FeSO_4 \cdot 7H_2O$ (20, 40, 80, 160, and 240 mg/L), and soybean meal (5, 10, 20, 40, and 60 g/L) were optimized. The 4′-N-demethyl-vicenistatin concentration at 7 d was used as the response.

#### 4.2.2. Searching for the Significant Medium Components by Plackett–Burman Design (PBD)

PBD was utilized for searching for the most significant medium components influencing 4′-N-demethyl-vicenistatin production by *S. parvus* SCSIO Mla-L010/Δ*vicG*. A total of six medium components, determined previously using OFAT, were investigated as factors (X), and the average value of the 4′-N-demethyl-vicenistatin concentration was considered as the response (Y). All of the factors were applied to design the PBD experiments with characteristics of the 3-level factorial designs (−1, 0, and +1), as shown in Table 1.

To screen the suitable concentrations of the main influencing factors (cassava starch, glycerol, and seawater salt), steepest ascent design (Table 2) was used based on the regression equation (first-order model, Equation (3)) obtained from the results of the PBD. More specifically, if the constant coefficient of the factor is positive, its concentration should increase, and vice versa.

$$Y = \beta_0 + \sum_{i=1}^{k} \beta_i X_i, \tag{3}$$

where $Y$ is the 4′-N-demethyl-vicenistatin concentration, $\beta_0$ and $\beta_i$ are coefficients, and $X_i$ represents the factors.

#### 4.2.3. Modeling and Optimization of the Medium Compositions by Response Surface Methodology (RSM)

RSM was used to investigate the effects of and complicated relationship between the significant medium components on 4′-N-demethyl-vicenistatin production. The central composite design (CCD) with 20 experimental runs used in this study is shown in Table 3. The second-order polynomial model (Equation (4)) was employed to fit the experimental data using Minitab 2020.

$$Y = \beta_0 + \sum_{i=1}^{k} \beta_i X_i + \sum_{i=1}^{k} \beta_{ii} X_i^2 + \sum_{i=1}^{k} \beta_{ij} X_i X_j, \tag{4}$$

where $Y$ is the predicted 4′-N-demethyl-vicenistatin concentration, $\beta_0$ is the intercept term, $\beta_i$ is the linear coefficient, $\beta_{ii}$ is the squared coefficient, $\beta_{ij}$ is the interaction coefficient, and $X_i$ and $X_j$ are factors.

#### 4.2.4. Modeling and Optimization of the Medium Compositions by Artificial-Neural-Network-Genetic-Algorithm (ANN-GA)

Alternatively, ANN was also adopted to fit the CCD experimental data because of its powerful features in modeling complex nonlinear relationships. To further explore information from the CCD experimental data and enhance the effect of the ANN training process, k-fold cross-validation was applied. The k is commonly set to 10 for the small sample dataset. In this way, all experimental samples were split into two datasets, as follows: training dataset and testing dataset, with a ratio of 9:1. As shown in Figure 3A, k-fold cross-validation starts by randomly splitting the data into k sets. For each set, one of the folds is selected for validation, and the remaining k-1 folds are used for training.

In this study, a three-layered feed-forward neural network was used (Figure 3B). The epochs, lr (learn rate), and goal were set at 500, 0.1, and $1 \times 10^{-6}$, respectively. The optimum number of neurons in the hidden layer, training algorithm, and activation function were determined by the minimum MSE (mean squared error) value (Table 5). MATLAB R2020a software was utilized to establish and optimize the model. The genetic algorithm (GA) parameters used in the optimization process were defined as follows: population size of 50, mutation probability of 0.1, crossover probability of 0.8, and iteration times of 500. Furthermore, the trained ANN model was used as the fitness function for GA.

#### 4.3. Analytical Method

To obtain the target product (4′-N-demethyl-vicenistatin), 200 μL harvested fermentation broth was extracted with 800 μL EtOAc (ethyl acetate) by mixing for 20 s. Then, the samples were pretreated with an ultrasonic wave for 10 min and mixed for 20 s, followed

by centrifugation at 12,000 rpm for 1 min and concentration with a vacuum centrifugal concentrator. Finally, the concentration of 4′-N-demethyl-vicenistatin was determined by HPLC (Shimadzu Prominence LC-20A, JP) using an C18 column (5 μm, 4.6 mm × 150 mm, ZORBAX 300SB, Agilent) with a UV detector at 254 nm. The gradient elution procedure was as follows: 0–20 min, 5–80% (*v/v*) B; 20–21.5 min, 80–100% B; 21.5–27.0 min, 100% B, 27.0–27.5 min, 100–5% B, 27.5–30 min, 5% B. The mobile phase consisted of 0.1% trifluoroacetic acid–water (A) and 0.1% trifluoroacetic acid–acetonitrile (B). All of the experiments were performed three times.

The ANOVA method was utilized to examine the significance of each independent variable. The performance of the RSM and ANN-GA models was compared based on various metrics, including error (Equation (5)), determination coefficient ($R^2$) (Equation (6)), *MSE* (Equation (7)), root mean square error (*RMSE*) (Equation (8)), and average percentage deviation (*APD*) (Equation (9)) [39].

$$error = \frac{|y_e - y_p|}{y_p}, \tag{5}$$

$$R^2 = \left[1 - \frac{\sum_{n=1}^{m}(y_{en} - y_{pn})^2}{\sum_{n=1}^{m}(y_{pn} - y_{e,ave})^2}\right], \tag{6}$$

$$MSE = \frac{\sum_{n=1}^{m}(y_{en} - y_{pn})^2}{m}, \tag{7}$$

$$RMSE = \sqrt{\frac{\sum_{n=1}^{m}(y_{en} - y_{pn})^2}{m}}, \tag{8}$$

$$APD = \frac{100 \times \sum_{n=1}^{m}\left(\left|\frac{y_{en} - y_{pn}}{y_{en}}\right|\right)}{m}, \tag{9}$$

where $y_p$ and $y_{pi}$ represent the predicted 4′-N-demethyl-vicenistatin concentration; $y_e$ and $y_{ei}$ denote the experimental concentration of 4′-N-demethyl-vicenistatin; $y_e$, *ave* represents the average value of the experimental concentration of 4′-N-demethyl-vicenistatin; and $n = 1, 2, 3. ….. m$, with *m* being the number of runs in the dataset.

## 5. Conclusions

In this study, six components in the fermentation medium of *S. parvus* SCSIO Mla-L010/Δ*vicG* were first screened by OFAT, and the significant medium components and their concentrations (11 g/L cassava starch, 18 g/L glycerol, and 27.5 g/L seawater salt) were determined using PBD and SAD. To achieve the maximal 4′-N-demethyl-vicenistatin production, ANN-GA (12 g/L cassava starch, 17 g/L glycerol, and 34 g/L seawater salt) and RSM (4 g/L cassava starch, 22 g/L glycerol, and 19 g/L seawater salt) were employed and thoroughly compared. The results have demonstrated that the ANN-GA model is a more reliable tool for the optimization and accurate prediction of the fermentation medium components for efficient 4′-N-demethyl-vicenistatin. Consequently, 0.1921 g/L 4′-N-demethyl-vicenistatin was obtained using the ANN-GA-optimized medium, which increased by 17% and 283% compared to the RSM-optimized and initial mediums, respectively.

**Author Contributions:** Conceptualization, methodology, writing—original draft preparation, Z.Y. and H.F.; formal analysis, H.F. and J.W.; writing—review and editing, Z.Y. and H.F.; supervision, H.F. and J.W.; funding acquisition and project administration, H.F. and J.W. All authors have read and agreed to the published version of the manuscript.

**Funding:** This work was supported by the Key-Area Research and Development Program of Guangdong Province (2020B1111030005).

**Institutional Review Board Statement:** Not applicable.

**Informed Consent Statement:** Not applicable.

**Data Availability Statement:** The data presented in this study are available in this article.

**Acknowledgments:** We are thankful to Zhicheng Liang (South China Sea Institute of Oceanology, Chinese Academy of Sciences) for providing technical support.

**Conflicts of Interest:** The authors declare no conflict of interest.

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
