# Peer review of "Modeling and Optimization of the Culture Medium for Efficient 4′-N-Demethyl-Vicenistatin Production by Streptomyces parvus Using Response Surface Methodology and Artificial-Neural-Network-Genetic-Algorithm"

_fermentation, doi:10.3390/fermentation10030154_

Round 1
Reviewer 1 Report
Comments and Suggestions for Authors
Title: Modeling and optimization of the culture medium for efficient 4'-N-demethyl-vicenistatin production by Streptomyces parvus using response surface methodology and artificial neural network-genetic algorithm
Authors: Yu, Fu and Wang
Summary: This study contributes to precision fermentation by defining an optimum medium to produce an antibiotic, 4'-N-demethyl-vicenistatin using Streptomyces parvus through process modelling and optimization. To determine the optimum medium, individual components were screened separately, modelled and simulated using the Plackett-Burman design, Response Surface Methodology and an Artificial Neural Network algorithm respectively. 17 g/L glycerol, 12 g/L cassava starch and 34 g/L seawater salt were identified as essential medium components for the optimum production of 0.1921 g/L 4'-N-demethyl-vicenistatin.
Overall, the paper is well written and should be accepted pending clarification on some of the points raised in the comments.
Comments
1. Have the authors considered the ease with which the product can be recovered from the fermentation medium?
2. There were no tests to track the production of 4'-N-demethyl-vicenistatin. A time series analysis of the accumulation of 4'-N-demethyl-vicenistatin would have been helpful.
Comments on the Quality of English LanguageEnglish editing is not required for the manuscript.
Author Response
March 2, 2024
Dear Editor,
Thank you very much for your letter and the comments from the reviewers about our submitted paper, “Modeling and optimization of the culture medium for efficient 4'-N-demethyl-vicenistatin production by Streptomyces parvus using response surface methodology and artificial neural network-genetic algorithm” (Manuscript Number: fermentation-2885091). We have carefully explained the reviewer's and editor’s questions and revised our manuscript with red font or yellow highlight. The following is a detailed list of response to comments. If you have any questions about this paper, please don’t hesitate to let me know.
Thank you very much for your consideration.
Most sincerely,
Hongxin Fu
Reviewer #1:
Comment 1: Have the authors considered the ease with which the product can be recovered from the fermentation medium?
Response: The separation method has been reported by previous research (J. Nat. Prod. 2022, 85, 256-263, doi: 10.1021/acs.jnatprod.1c01044). The purification method for this reference is as follows:
- parvus SCSIO Mla-L010 and its ΔvicG mutant strain were cultured onto MS plates and incubated at 28 °C for 7 d to collect mature spores. The fermentation was conducted using M-AM3 medium as a seed medium. The seed medium (50 mL) was inoculated at 28 °C on a rotary shaker at 200 rpm for 48 h, then transferred into 1 L Erlenmeyer flasks containing 200 mL of M-AM3 medium. After being incubated at 28 °C on the rotary shaker for 7 d, the culture (20 L) was centrifuged at 4000 rpm for 10 min to obtain the supernatant and mycelium. The supernatant was extracted with equal volumes of EtOAc three times, and the mycelia cake was extracted three times with 2 L of acetone. All extract solutions were combined, then evaporated to dryness.
The method utilized in this study was optimized based on it.
Comment 2: There were no tests to track the production of 4'-N-demethyl-vicenistatin. A time series analysis of the accumulation of 4'-N-demethyl-vicenistatin would have been helpful.
Response: Thanks for your professional suggestion. We have conducted the fermentation experiments to investigate the fermentation kinetics of S. parvus SCSIO Mla‑L010/ΔvicG in three different media. Figure 1 demonstrated that both strain growth and product synthesis are significantly influenced by the medium components. And the ANN-GA model exhibited an optimal result, although the detailed mechanism needs further study.
![]()
|
![]()
|
![]()
|
(a) |
(b) |
(c) |
Figure 1. The fermentation kinetics of 4'-N-demethyl-vicenistatin. (a) AM3; (b) RSM; (c) ANN-GA.

Reviewer 2 Report
Comments and Suggestions for Authors
Review for
Modeling and optimization of the culture medium for efficient 4'-N-demethyl-vicenistatin production by Streptomyces parvus using response surface methodology and artificial neural network-genetic algorithm
By Zhixin Yu 1, Hongxin Fu 1, 2, 3,* and Jufang Wang 1, 2, 3,*
vicenistatin
is an interesting compound under scrutiny by many scientists
Semi-synthesis and structure-activity relationship study yield antibacterial vicenistatin derivatives with low cytotoxicity
Li, J., Yang, Z., Shi, C., ...Li, Q., Ju, J.
Journal of Antibiotics, 2024
Structural Basis of Amide-Forming Adenylation Enzyme VinM in Vicenistatin Biosynthesis
Miyanaga, A., Nagata, K., Nakajima, J., ...Kudo, F., Eguchi, T.
ACS Chemical Biology, 2023, 18(11), pp. 2343–2348
Protein-Protein Recognition Involved in the Intermodular Transacylation Reaction in Modular Polyketide Synthase in the Biosynthesis of Vicenistatin
Chisuga, T., Miyanaga, A., Eguchi, T.
ChemBioChem, 2022, 23(14), e202200200
--------------------
What is the position of 4'-N-demethyl-vicenistatin
among antibacterial vicenistatin derivatives with low cytotoxicity
even if your work is more antitumor oriented
4'-N-demethyl-vicenistatin is a vicenistatin analogue which has better antitumor activity with promising applications in pharmaceuticals industry
-------------------------------
Cassava starch, glycerol, and seawater salt were identified as the pivotal components influencing 4'-N-demethyl-vicenistatin production.
why using such ‘complex’ or ‘non-stable’ components in an optimization medium?
cassava starch?
seawater salt?
----------------------------
The ANN-GA model showed superior reliability, achieving the highest 4'-N-demethyl-vicenistatin at 0.1921 g/L, which was 17% and 283% higher compared to the RSM-optimized and initial medium approaches, respectively
0.1921 g/L
what is the maximum concentrations described by other authors?
may be with different strains?
-------------------------------------
what are the current g/L of similar products, at an industrial scale?
---------------------------
Vicenistatin, a novel 20-membered macrocyclic lactamantitumor antibiotic
Shindo, K., Kamishohara, M., Odagawa, A., Matsuoka, M., Kawai, H.
Journal of Antibiotics, 1993, 46(7), pp. 1076–1081
first vicenistatin described more than 30 years ago
current industrial production?
Author Response
March 2, 2024
Dear Editor,
Thank you very much for your letter and the comments from the reviewers about our submitted paper, “Modeling and optimization of the culture medium for efficient 4'-N-demethyl-vicenistatin production by Streptomyces parvus using response surface methodology and artificial neural network-genetic algorithm” (Manuscript Number: fermentation-2885091). We have carefully explained the reviewer's and editor’s questions and revised our manuscript with red font or yellow highlight. The following is a detailed list of response to comments. If you have any questions about this paper, please don’t hesitate to let me know.
Thank you very much for your consideration.
Most sincerely,
Hongxin Fu
Reviewer #2:
Comment 1:
Vicenistatin is an interesting compound under scrutiny by many scientists. What is the position of 4'-N-demethyl-vicenistatin among antibacterial vicenistatin derivatives with low cytotoxicity, even if your work is more antitumor oriented? 4'-N-demethyl-vicenistatin is a vicenistatin analogue which has better antitumor activity with promising applications in pharmaceuticals industry.
Semi-synthesis and structure-activity relationship study yield antibacterial vicenistatin derivatives with low cytotoxicity. Li, J., Yang, Z., Shi, C., ...Li, Q., Ju, J. Journal of Antibiotics, 2024
Response: Thanks for your professional advice. Li et al. conducted a comprehensive comparative analysis of vicenistatin and its derivatives, providing detailed structural information in Figures 1 and 2, where each compound is assigned a unique serial number. Furthermore, the article presents the cytotoxicity (Table 2) and antibacterial activity (Table 1) of each substance. Notably, this study reveals that 4'-N-demethyl vicenistatin exhibits good antimicrobial activities and low cytotoxicity (J Antibiot. 2024. doi: 10.1038/s41429-023-00701-3).
Comment 2: Cassava starch, glycerol, and seawater salt were identified as the pivotal components influencing 4'-N-demethyl-vicenistatin production. Why using such ‘complex’ or ‘non-stable’ components in an optimization medium? cassava starch? seawater salt?
Response: Cassava starch and seawater salt used in this study are food grade, which must meet the requirements of China National Standards. Therefore, their contents are relatively stable. The reason why we selected cassava starch and seawater salt is because they obtained the best results in the components screening tests (Fig. 1) and they are commonly used components in previous research (Microorganisms 2022, 10(9), 1854; Microorganisms 2022, 10(1):94; Preparative Biochemistry & Biotechnology 2022, 52, 525-533; 3 Biotech. 2019, 9(12):446).
Comment 3: The ANN-GA model showed superior reliability, achieving the highest 4'-N-demethyl-vicenistatin at 0.1921 g/L, which was 17% and 283% higher compared to the RSM-optimized and initial medium approaches, respectively. What is the maximum concentrations described by other authors? May be with different strains?
Response: As far as we know, the studies on vicenistain and its derivatives primarily focus on elucidating the mechanisms underlying microbial synthesis, drug action, and drug derivatization. There are no reports about industrialization of vicenistain and its derivatives. To date, only one publication has been reported about 4'-N-demethyl-vicenistatin production, with a low concentration of 0.004 g/L (J. Nat. Prod. 2022, 85, 256-263, doi: 10.1021/acs.jnatprod.1c01044).
Comment 4: What are the current g/L of similar products, at an industrial scale?
Response: As far as we know, the studies on vicenistain and its derivatives primarily focus on elucidating the mechanisms underlying microbial synthesis, drug action, and drug derivatization. There are no reports about industrialization of vicenistain and its derivatives.
Comment 5: First vicenistatin described more than 30 years ago. Current industrial production?
Response: As far as we know, the studies on vicenistain and its derivatives primarily focus on elucidating the mechanisms underlying microbial synthesis, drug action, and drug derivatization. There are no reports about industrialization of vicenistain and its derivatives.
